

# Valence makes a stronger contribution than arousal to affective priming

Zhao Yao[1,2], Xiangru Zhu[3,4] and Wenbo Luo[5]

[1] Research Center of Shaanxi intelligence society development, Xidian University, Xi'an, Shaanxi, China
[2] School of Humanities, Xidian University, Xi'an, Shaanxi, China
[3] Institute of Cognition, Brain and Health, Henan University, Kaifeng, Henan, China
[4] Institute of Psychology and Behavior, Henan University, Kaifeng, Henan, China
[5] Research Center of Brain and Cognitive Neuroscience, Liaoning Normal University, Dalian, Liaoning, China

## ABSTRACT

**Background**. Recent data suggest that both word valence and arousal modulate subsequent cognitive processing. However, whether valence or arousal makes a stronger contribution to cognitive processing is less understood.

**Methods**. The present study performed three experiments that varied the valence (positive or negative) and arousal (high or low) of prime-target word pairs in a lexical decision-priming task. Affective priming was derived from pure valence (Experiment 1), pure arousal (Experiment 2), or a combination of valence and arousal (Experiment 3).

**Results**. By comparing three types of priming effects, we found an effect of valence on affective priming was obvious regardless of whether the relationship of the prime-target varied with valence, arousal, or the combination of valence and arousal. In contrast, an effect of arousal on affective priming only appeared in the condition that based on the arousal relationship of the prime-target pair. Moreover, the valence-driven priming effect, arousal-driven priming effect, and emotional-driven priming effect were modulated by valence type but not by arousal level of word stimuli.

**Conclusion**. The present results revealed a pattern of valence and arousal in semantic networks, indicating that the valence information of emotional words tends to be more stable than arousal information within the semantic system, at least in the present lexical decision-priming task.

Corresponding authors
Xiangru Zhu, zhuxiangru@gmail.com
Wenbo Luo, luowb@lnnu.edu.cn

## INTRODUCTION

Emotions influence our everyday lives in several ways. In a continuous flow of information, we must focus, select, store, and retrieve relevant information. Amazingly, we are able to automatically evaluate the affective value of all incoming stimulus information within a few milliseconds. Evaluation that is associated with affect and feelings helps people initiate subsequent appropriate behavioral responses. Although the processing of affective information has traditionally been assessed using self-reports, in which participants explicitly evaluate the affective information of stimuli, self-reports have inherent limitations. For example, if participants do not want to report their true feelings

or deliberate on some stimuli, they might misreport their attitude and affective evaluations (*Herring et al., 2013*).

To minimize the bias of self-report evaluations, *Fazio (2001)* developed an affective priming paradigm to investigate automatic stimulus evaluation. In a typical affective priming study, an affective/evaluative relation between the prime and target is manipulated, and participants are asked to respond on the basis of a particular feature (e.g., emotional or non-emotional) of the target stimuli. A common finding in this paradigm is that performance is typically faster and more accurate when a prime and target are congruent and have the same emotional information (e.g., "*flower*"–"*wedding*") compared with when they are incongruent and have different emotional information (e.g., "*party*'–"*corpse*"). This performance has been called an affective priming effect (for reviews, see *Fazio, 2001*; *Klauer & Musch, 2003*). Crucially, such an effect shows that the emotional information of stimuli can be implicitly and automatically evaluated. The affective priming effect can typically be explained by hypotheses of the spreading of activation (*Fazio, 2001*) or response competition (*Spruyt et al., 2007*). Throughout the history of investigations of affective priming, many studies have investigated the effects of valence and arousal on affective priming in various cognitive tasks because valence and arousal were identified as the most basic dimensions of emotional information (*Russell, 2003*). What is still unclear, however, is whether valence and arousal have similar effects on target processing.

Much experimental evidence has revealed the influence of primed valence and arousal on affective priming (*Zhang et al., 2006*; *Zhang, Kong & Jiang, 2012*; *Herring et al., 2011*; *Herring et al., 2015*; *Hinojosa et al., 2009*; *Hinojosa, Méndez-Bértolo & Pozo, 2012*). For instance, a series of studies by *Zhang et al. (2006)* and *Zhang, Kong & Jiang (2012)* suggested that the participants responded faster in valence (arousal) congruent trials than in valence (arousal)-incongruent trials in a valence decision task. Moreover, previous studies suggest that the affective priming effect of positive primes differs from negative primes (e.g., *Pan et al., 2016*; *Kissler & Koessler, 2011*; *Spruyt et al., 2007*; *Rossell & Nobre, 2004*; *Aguado et al., 2018*). For example, *Yao & Wang (2013)* reported that a significant affective priming effect was observed with positive primes but not with negative primes in a lexical decision-priming task.

Although much is known about the roles of valence and arousal in affective priming, but less is known about whether valence and arousal have the same "power" to trigger affective priming. Three studies have systematically manipulated the valence (positive, negative) and arousal (high, low) of primes and targets (*Herring et al., 2015*; *Zhang, Kong & Jiang, 2012*), but their findings were inconsistent. Apart from differences in experimental stimuli (e.g., pictures-words, pictures-pictures, and faces-faces) and prime position (foveal vs. parafoveal), we inferred that such inconsistencies might arise from differences between judgment tasks that encouraged participants to selectively assign attention to valence and/or arousal, or neither. In previous studies, the participants were asked to complete a valence decision task or an arousal decision task (*Zhang, Kong & Jiang, 2012*; *Herring et al., 2015*). These two types of tasks require explicit processing of the valence or arousal of stimuli, with selective attention directed toward a specific affective feature. Spruyt and colleagues provided evidence that task-dependency of the affective priming effect can be

modulated by feature-specific attention allocation (*Spruyt et al., 2007*; *Spruyt et al., 2012*). Therefore, to determine whether valance and arousal play similar roles in affective priming and to avoid attention being assigned to a specific affective dimension, we employed a lexical decision-priming task, in which the participants were asked to judge whether the target is a real word as quickly and accurately as possible.

We used a lexical decision-priming task for two reasons. First, this task does not require participants to have an explicit evaluative processing goal (e.g., to process either the valence or arousal of stimuli). Therefore, all of the words are explicitly processed in the same way, and differences in performance between words that differ in valence or arousal will be attributable to their emotional dimension *per se*, without additional influence from top-down, task-dependent processes (*Citron, 2012*). The contribution of valence and arousal to affective priming would be relatively equally accessible in the lexical decision-priming task compared with the evaluative decision task. Second, although the evaluative decision task is most widely employed in the relevant literature (for review, see *Herring et al., 2013*), affective priming effects have been observed in non-evaluative tasks, such as the lexical decision task (e.g., *Kissler & Koessler, 2011*; *Yao & Wang, 2013*; *Yao & Wang, 2014*; *Hinojosa, Méndez-Bértolo & Pozo, 2012*) and in the naming/pronunciation task (e.g., *Spruyt et al., 2007*). Some researchers suggested that a minimum degree of linguistic processing modulates the allocation of attentional resources to the affective feature of words, even in a more superficial structural task. In the lexical decision-priming task, participants need to access a word's meaning when they judge that the target is a real word or a pseudoword, which allows their attention to be divided across various stimulus dimensions, including the inherently important affective feature (*Spruyt et al., 2007*; *Citron, 2012*). Therefore, to obtain and compare pure valence-driven or pure arousal-driven priming effects, we adopted the lexical decision-priming task, which produces a neutral processing mindset and allows valence and arousal to influence affective priming as equally as possible.

We assume that valence and arousal do not produce parallel priming effects, even when they are equally handled in an affective priming task. First, previous studies have provided evidence that valence and arousal are associated with different physiological and affective responses and activate partially dissociable brain networks (*Estes & Adelman, 2008*; *Delaney-Busch, Wilkie & Kuperberg, 2016*). For example, an event-related potential study explored the time course of brain electric activity evoked by information about the valence and arousal of emotional words and found first valence then arousal (*Gianotti et al., 2008*). Second, some evidence has shown that the valence of a stimulus is associated with higher-order cognitive, whereas arousal is associated with more automatic physiological reactions that are less cognitively accessible (*Nicolle & Goel, 2013*; *Citron, 2012*). *Nicolle & Goel (2013)* asked participants to rate sentences along cognitive dimensions (i.e., believable, unbelievable) and affective dimensions (i.e., valence, arousal) and suggested that valence responses may require internal computations and are more likely to be influenced by our beliefs, whereas arousal-related responses do not rely on evaluation of the stimulus and may be considered to occur at a more stimulus-driven level. Third, rating studies reported the high variability of arousal ratings compared with valence ratings (e.g., *Yao et al., 2017*). Arousal may show bigger differences between individuals. That is, the same valence word

may activate different degree of physiological activation for different groups of people. Take the word "*wedding*", it was rated as positive valence and higher arousal due to more excited feeling for people in love, but for others, it may elicit less excited feelings, but almost no one regard as the word "*wedding*" is negative valence. Given these findings, we reasoned that valence and arousal may have differential effects on affective priming because of their unequal ability to engage emotional processing.

Taken together, the aim of the present study was to examine whether valence and arousal have the same priming "power" in modulating affective priming in the lexical decision-priming task. We manipulated the affective relationship between primes and targets in terms of valence (positive, negative) and arousal (high, low). In Experiment 1, valence was manipulated so that the prime-target word pairs were valence-congruent (positive-positive, negative-negative) or valence-incongruent (positive-negative, negative-positive). The level of arousal of the pairs (low or high) was equated in four prime-target conditions. In Experiment 2, arousal of the prime-target word pairs was manipulated so that word pairs were arousal-congruent (high-high, low-low) or arousal-incongruent (high-low, low-high) but were equated with regard to valence (positive or negative) in four prime-target conditions. In Experiment 3, we used emotional words as primes and neutral/emotional words as targets. Half of the word pairs were emotionally congruent (emotional words-emotional words; i.e., the prime and target had similar valence and arousal), and the other half were emotionally incongruent (emotional words-neutral words; i.e., the prime and target differed in both valence and arousal). Therefore, affective priming in Experiment 1 reflected valence priming effect, which refers to a facilitated response to a target when it is preceded by a valence-congruent prime compared with a valence-incongruent prime. Affective priming in Experiments 2 and 3 reflected arousal priming effect (i.e., faster responses in arousal-congruent trials than in arousal-incongruent trials within the same valence) and emotional priming effect (i.e., faster responses in emotionally congruent trials than in emotionally incongruent trials), respectively.

By comparing the priming effects of valence, arousal, and the combination of valence and arousal on subsequent target processing, we could determine whether valence and arousal have the same affective priming strength. And then, we sought to determine whether valence or arousal who play a more relatively stable role in affective priming. If valence plays a stronger role in affective priming, then valence priming effects may still be evident when the arousal and emotional dimensions of primes-targets are manipulated. Conversely, if arousal plays a stronger role in affective priming, then arousal priming effects may still be evident when the valence and emotional dimensions of primes-targets are manipulated.

# EXPERIMENT 1: AFFECTIVE PRIMING WAS DERIVED FROM PURE VALENCE RELATIONSHIP OF PRIMES AND TARGETS

## Method

### Participants

Twenty-nine native Chinese speakers (17 men, 12 women; age in years: $M = 18.2$, $SD = 1.8$, range = 17–21) participated in Experiment 1. They were all right-handed with

**Table 1  Means of valence (1, negative to 9, positive), arousal (1, calming to 9, arousing), concreteness (1, abstract to 9, concrete), word frequency and strokes.**

|  | Valence | Arousal | Concreteness | Word frequency | Strokes |
|---|---|---|---|---|---|
| High-arousal positive words (PH) | 7.30 (.48) | 6.92 (.28) | 5.39 (1.63) | 32.76 (27.04) | 18.27 (4.70) |
| High-arousal negative words (NH) | 2.41 (.36) | 6.96 (.50) | 5.13 (1.64) | 25.45 (24.45) | 16.27(3.30) |
| Low-arousal positive words (PL) | 6.61 (.25) | 5.81 (.29) | 5.22 (1.49) | 22.28 (12.98) | 17.20 (4.88) |
| Low-arousal negative words (NL) | 3.07 (.44) | 5.62 (.41) | 5.18 (1.51) | 27.39 (16.01) | 17.30 (4.28) |
| Neutral target words (in Experiment 3) | 4.90 (.40) | 4.44 (.68) | 5.19 (1.60) | 38.17 (43.27) | 16.91 (4.22) |

normal or corrected to normal vision. None had any history of neurological or psychiatric disorders. They all gave informed consent before the experiment. The study was approved by the local Ethics Committee of Henan University (HUSOM-2018-102).

### Stimuli

All of the stimuli that were presented were selected from a database of 1100 Chinese two-character words (*Yao et al., 2017*). This database provides mean ratings and standard deviations (SDs) of valence, arousal, and concreteness for each word. Each word was rated by at least 48 participants using a 9-point scale. We considered words with valence values that ranged from 1 to 4 as negative and words with values that ranged from 6 to 9 as positive. Words with arousal values greater than 6.5 were considered high arousal words. Words with arousal values that ranged from 5 to 6 were considered as low arousal words.

From this database, the final stimulus set included 120 experimental nouns, 30 high-arousal positive (PH; e.g., *gold medal, miracle*), 30 low-arousal positive (PL; e.g., *flowers, serenity*), 30 high-arousal negative (NH; e.g., *renegade, hatred*) and 30 low-arousal negative words (NL; e.g., *scar, lengthiness*). Four types of words were matched with regard to word frequency ($F_{3,116} = 1.03, p = .38$), concreteness ($F_{3,116} = .16, p = .92$), and character strokes ($F_{3,116} = 1.07, p = .37$). Positive and negative words had comparable arousal ($F_{1,118} = 0.26, p = .61$) but differed in valence ($F_{1,118} = 19.83, p < .001$). High- and low-arousal words had comparable valence ($F_{1,118} = 0.03, p = .96$) but differed in arousal ($F_{1,118} = 24.631, p < .001$). Word characteristics (means and SDs) are presented in Table 1. A further paired sample *t*-test (2-tailed, more details are listed in Table 2) comparing the four condition types revealed that there was no significant difference in valence rating between PH and PL, NH and NL, also in arousal rating between PH and NH, PL and NL.

All 240 prime-target word pairs were constructed based on the four types of words. There were 120 valence-congruent pairs (30 PL-PL, 30 PH-PH, 30 NL-NL, 30 NH-NH) and 120 valence-incongruent pairs (30 PL-NL, 30 PH-NH, 30 NL-PL, 30 NH-PH). Moreover, a total of 240 trials were conducted that consisted of PL/PH/NL/NH prime-pseudoword target (60 trials in each pair). In order to create the pseudowords, based on the 120 experimental words, one character was randomly changed in each word to produce pronounceable pseudowords. During the experimental session, the four types of experimental words as primes and targets were repeated four and two times, respectively, and pseudowords were presented two times. Moreover, we carefully excluded possible semantic relationship of primes-targets by an additional rating experiment using a 5-point scale (from 1 point to 5

**Table 2 Paired sample *t*-tests (2-tailed) comparing the four types of experimental words.**

| | PH | PL | NH | NL |
|---|---|---|---|---|
| **Valence** | | | | |
| PH | —— | | | |
| PL | $t = 1.87, p = .08$ | —— | | |
| NH | $t = 54.24, p < .001$ | $t = 57.21, p < .001$ | —— | |
| NL | $t = 31.47, p < .001$ | $t = 33.35, p < .001$ | $t = 1.66, p = .11$ | —— |
| **Arousal** | | | | |
| PH | —— | | | |
| PL | $t = 12.03, p < .001$ | —— | | |
| NH | $t = -0.32, p = .75$ | $t = -10.09, p < .001$ | —— | |
| NL | $t = 11.99, p < .001$ | $t = 1.34, p = .19$ | $t = 10.32, p < .001$ | —— |

**Notes.**

PH, High-arousal positive words; PL, Low-arousal positive words; NH, High-arousal negative words; NL, Low-arousal negative words.

point, a higher score indicates a higher level of semantic relatedness) with another sample of 12 participants, ensuring that no word pairs had a high level of semantic relatedness (1.22–1.93).

### Task and procedure

All 480 trials were presented to every participant in four blocks of trials. The four blocks were randomized across participants and began with PL primes, PH primes, NL primes, and NH primes, respectively, consisting of a total of 120 trials (30 valence-congruent trials, 30 valence-incongruent trials, and 60 PL/PH/NL/NH-pseudowords trials). Within each block, the sequence of trials was pseudorandom, with the constraints that identical trials were not repeated two times in a block. A meta-analysis of a quarter century of affective priming studies revealed that separated, compared to intermixed stimulus sets, produce enhanced affective priming (*Herring et al., 2013*). From an attentional sensitization perspective, having stimulus sets remain constant in a block can enable richer processing of the emotional dimension of stimuli via reducing stimulus variability (also see *Herring et al., 2015*).

Participants were tested individually and were instructed to answer as quickly and accurately as possible. Stimuli and instructions were presented in a white font on a black background. Target words had to be judged as a real word or a pseudoword by pressing the "Z" and "D" keys on the keyboard (assignment of the two keys to response categories was counterbalanced across participants). Instructions emphasized that the first word appearing in each trial (prime) were silently read, and the second word (target) had to be responded in each trial.

Each trial started with the presentation of a fixation cross for 300 ms, followed by a prime word for 200 ms. After the prime, a blank screen was shown for 100 ms before the target was presented until the participant responded or 1500 ms elapsed. After an inter-trial interval of 1000 ms, the next trial started. In order to participants completely understood the trial procedure, they were given 14 practice trials and an additional feedback screen

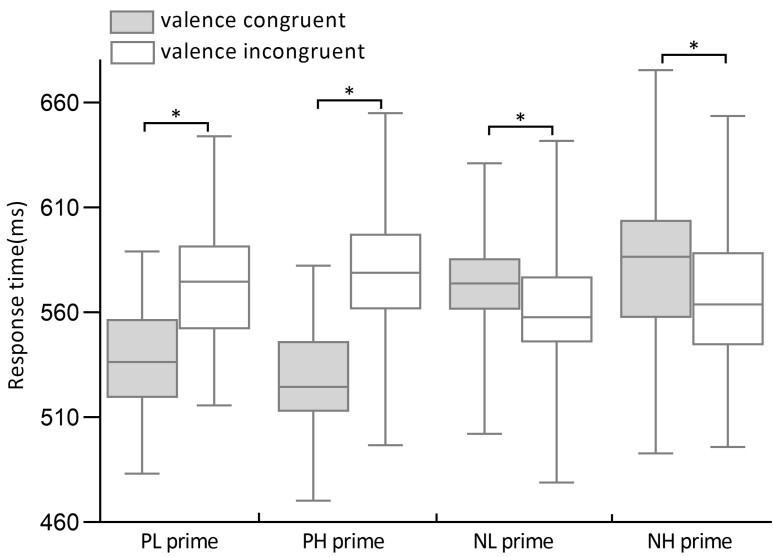

**Figure 1** **Response times of high- and low- arousal positive/negative primes in valence congruenceand incongruence conditions.** The gray column indicates the response times of prime-target valence congruence condition and the gray slash indicates the response times of prime-target valence incongruence condition.

after erroneous or slow responses before beginning the experiment. After each block, the participants were allowed to have short pauses between two blocks.

This task was presented via E-prime 2.0 software (Psychology Software Tools Inc., Sharpsburg, PA, USA).

## Results

Overall accuracy was high (98.5%) and did not significant difference between experimental conditions (range: 96.9–99.4%). Response times (RTs) were 2.5 SDs above or below the mean of each participant were excluded from analysis (0.2% of the data). The reaction time (RT in ms) is reported only for correct responses. All calculations were conducted using an SPSS statistical package (version 18, SPSS inc., IBM company).

A repeated-measures ANOVA was run on response times in the eight prime-target conditions: prime valence (positive, negative) × valence congruency (congruent, incongruent) × arousal level of word pairs (high, low). The results revealed a significant main effect of prime valence ($F_{1,28} = 5.42$, $p = .03$, $\omega^2 = .13$), responses to negative primes (572.0 ± 4.4 ms) were slower than responses to positive primes (556.4 ± 5.8 ms). A significant main effect of valence congruency was significant ($F_{1,28} = 46.34$, $p < .001$, $\omega^2 = .61$), with longer response times in valence-incongruent trials (570.8 ± 4.0 ms) than in valence-congruent trials (557.6 ± 3.9 ms). There was a significant interaction between prime valence and valence congruency ($F_{1,28} = 57.86$, $p < .001$, $\omega^2 = .84$). Moreover, a significant three-way interaction was found between prime valence, arousal level of word pairs, and valence congruency ($F_{1,28} = 8.86$, $p = .006$, $\omega^2 = .21$; Fig. 1). The simple-effect analysis showed that response times in incongruent trials (574.6 ± 6.9 ms) were significantly longer

than in congruent trials (540.3 ± 5.7 ms) in PL primes ($F_{1,28} = 29.95$, $p < .001$, $d' = 1.06$). Similarly, response times in incongruent trials (581.8 ± 8.1 ms) were significantly longer than in congruent trials (529.0 ± 7.7 ms) in PH primes ($F_{1,28} = 33.03$, $p < .001$, $d' = 1.15$). Conversely, the participants responded faster in incongruent trials (558.3 ± 2.3 ms) than in congruent trials (572.8 ± 2.5 ms) in NL primes ($F_{1,28} = 23.89$, $p < .001$, $d' = .45$). In NH primes, the participants also responded faster in incongruent trials (568.5 ± 7.5 ms) than in congruent trials (588.2 ± 8.4 ms) ($F_{1,28} = 24.67$, $p < .001$, $d' = .48$). There was no significant main effect ($F_{1,28} = 1.14$, $p = .30$, $\omega^2 = .04$) and interactions of arousal level of word pairs (all $p > .05$).

## Discussion

Experiment 1 focused on the ways in which valence relationships between primes and targets influence affective priming. The results showed significant main effects of prime valence and valence congruency and significant interactions between prime valence, prime arousal, and valence congruency. In contrast, no main effect and related interactions of arousal level of word pairs were found to be significant. These results replicated previous studies, demonstrating that participants are quicker when responding to targets that are valence-congruent with primes compared with targets that are valence-incongruent with primes (*Zhang et al., 2006*; *Herring et al., 2011*). In Experiment 1, when the valence information of primes was evaluated implicitly, encoding and response mechanisms were activated by the same valence information of primes-targets, thus allowing valence-congruent targets more accessibility and allowing the participants to respond more easily relative to valence-incongruent targets (*Fazio, 2001*; *Klauer & Musch, 2003*). However, the valence-driven priming effect in Experiment 1 was inconsistent with attentional sensitization models (*Spruyt et al., 2007*; *Spruyt et al., 2012*). One possible reason is that we did not procedurally manipulate the participants' selective attention in stimulus processing, so that the participants could freely attend to certain affective features in the lexical decision-priming task.

Furthermore, the direction of priming effects is different between positive and negative primes, the participants responded faster in congruent trials than in incongruent trials in positive primes, instead, faster in incongruent trials than in congruent trials in negative primes. These results suggested that positive primes played a facilitative role in affective priming, opposite to the role of negative primes, which is in line with findings of previous studies (*Pan et al., 2016*; *Kissler & Koessler, 2011*; *Rossell & Nobre, 2004*; *Yao & Wang, 2013*). According to the affect-as-information approach, there is an associative network of memory and emotion where emotional information (i.e., valence and arousal) can be represented as nodes in a semantic network, in which accessibility and use of the associative network could be the cause for asymmetric effects of positive and negative primes (*Clore & Storbeck, 2006*).

According to the density hypothesis, one possible explanation for the facilitation of positive primes is the higher density of positive information compared with negative information in semantic memory. The density hypothesis states that positive information is generally more similar to other positive information compared with negative information's

similarity to other negative information. In visualizations of mental representations, positive information is thus more densely clustered (*Unkelbach et al., 2008*; *Koch et al., 2016*). This asymmetry of similarity might explain valence asymmetries at all levels of cognitive processing.

The opposite priming effect for negative primes in the present study can be explained by the automatic vigilance hypothesis of emotion that was proposed by *Estes & Adelman (2008)*. This model proposes that negative words may hold attention longer (i.e., delayed disengagement) than positive or neutral words in color naming, word naming, and lexical decision tasks, thus leading to slower responses to negative words (*Estes & Adelman, 2008*; *Larsen et al., 2008*). According to this hypothesis, attention disengages more slowly from negative prime-target trials than from negative prime-positive (neutral) target trials. The processing of a negative target when it is preceded by a negative prime may produce "double negative delay", making the accessibility and use of an affective association between negative primes and negative targets become reduced or restricted. As a result, the participants responded slower in negative prime-positive(neutral) target trials than in negative prime-negative target trials.

In Experiment 1, a pure valence-driven priming effect was observed when the valence relationship of primes-targets was manipulated in the lexical decision-priming task. In Experiment 2, we manipulated the arousal relationship of primes-targets and explored pure arousal-driven priming effects in the same task.

## EXPERIMENT 2: AFFECTIVE PRIMING WAS DERIVED FROM PURE AROUSAL RELATIONSHIP OF PRIMES AND TARGETS

### Method
#### *Participants*
Thirty-two university students (18 females; 17–22 years old, mean age = 19.5 years, $SD = 2.3$) who did not participate in Experiment 1 participated in Experiment 2 and received financial compensation for their participation (see Experiment 1 for further details). They all gave written informed consent before the experiment.

#### *Stimuli*
Although the four pools of stimuli and pseudowords were the same as in Experiment 1. There were 120 arousal-congruent pairs (30 PL-PL, 30 PH-PH, 30 NL-NL, 30 NH-NH) and 120 arousal-incongruent pairs (30 PL-PH, 30 PH-PL, 30 NL-NH, 30 NH-NL).

#### *Task and procedure*
The experimental task and procedure were the same as in Experiment 1.

### Results
Overall accuracy was high (98.3%) and did not significant difference between experimental conditions (range: 97.6–99.3%), and thus only correct response times was reported. Response times were 2.5 SDs above or below the mean of each participant were excluded

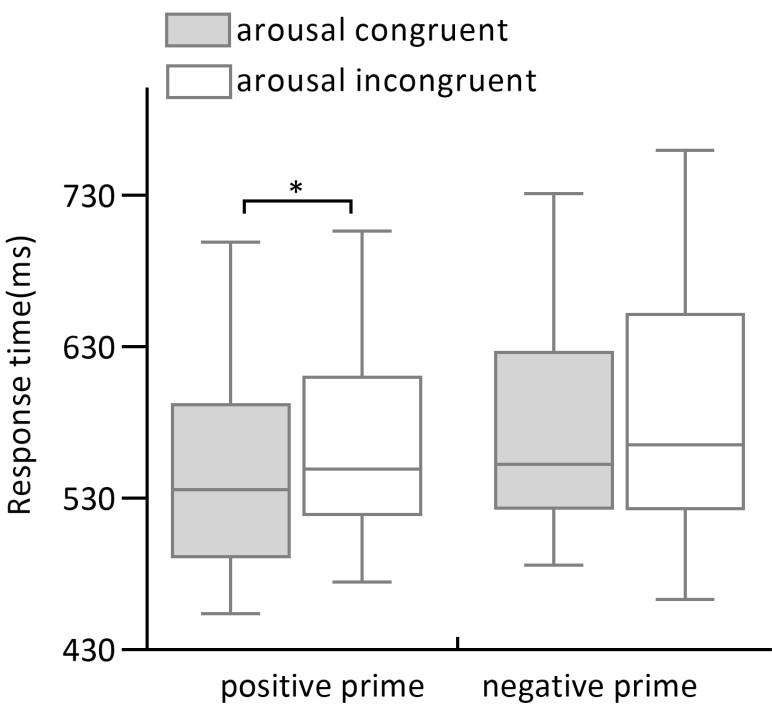

**Figure 2** **Response times of positive primes and negative primes in arousal congruence and incongruence conditions.** The gray column indicates the response times of prime-target arousal congruence condition and the gray slash indicates the response times of prime-target arousal incongruence condition.

from analysis (0.3% of the data). All calculations were conducted using an SPSS statistical package (version 18, SPSS inc.; IBM company, Armonk, NY, USA).

The 2 (prime arousal: high, low) ×2 (arousal congruency: congruent, incongruent) ×2 (valence type of word pairs: positive, negative) repeated-measures ANOVA of response times revealed a main effect of valence type of word pairs ($F_{1,31} = 47.21$, $p < .001$, $\omega^2 = .59$; 562.3 ± 14.1 ms for positive pairs and 584.6 ± 15.1 ms for negative pairs) and arousal congruency ($F_{1,31} = 33.41$, $p < .001$, $\omega^2 = .50$; 578.0 ± 15.1 ms for arousal-congruent pairs and 569.0 ± 14.8 ms for arousal-incongruent pairs). A significant arousal congruency × valence type of word pairs was found ($F_{1,31} = 11.35$, $p = .002$, $\omega^2 = .24$. see Fig. 2). The simple-effect analysis showed that response times in arousal-incongruent trials (578.6 ± 15.4 ms) were significantly longer than in arousal-congruent trials (546.11 ± 13.2 ms) for positive pairs ($F_{1,31} = 39.93$, $p < .001$, $d' = .39$). For negative pairs, no difference between arousal-congruent (580.5 ± 14.0 ms) and arousal-incongruent trials (588.8 ±16.5 ms) was observed ($F_{1,31} = 2.89$, $p = 0.1$, $d' = .10$). No significant main effect of prime arousal was observed ($F_{1,31} = 1.64$, $p = .21$, $\omega^2 = .02$), with no other arousal-related effects (all $p > .05$).

## Discussion

The goal of Experiment 2 was to investigate the effect of prime arousal on affective priming. In line with prior study by *Zhang, Kong & Jiang (2012)*, the results of Experiment 2 showed a significant arousal-driven priming effect only in positive word pairs. The participants

responded faster in positive arousal-congruent than arousal-incongruent trials, suggesting that the arousal information of prime stimuli is able to automatically capture attentional resources with a help of positive valence, thereby influencing the processing of target stimuli. One likely interpretation is that arousal cue of a prime has the ability but alone is insufficient to activate all other concepts of the same arousal in semantic network, and thus has to need a support of positive valence. Because positive information of prime stimuli facilitated the processing of subsequent related stimuli that has been demonstrated by numerous studies (e.g., *Kissler & Koessler, 2011*; *Yao & Wang, 2014*), but there have mixed findings of the effect of arousal on affective priming in both behavior and electrophysiology studies (*Herring et al., 2015*; *Zhang, Kong & Jiang, 2012*; *Hinojosa, Méndez-Bértolo & Pozo, 2012*; *Hinojosa et al., 2009*).

In Experiment 2, the prime-targets relationship was based on arousal, and a significant arousal priming effect was observed in positive pairs. In Experiment 3, we systematically varied the affective relationship of primes-targets along valence and arousal dimensions to explore pure emotional-driven priming effects in the lexical decision-priming task.

# EXPERIMENT 3: AFFECTIVE PRIMING WAS DERIVED FROM EMOTIONAL RELATIONSHIP OF PRIMES-TARGETS

## Method
### Participants
Thirty-one university students (13 females; age 18–25 years old; mean age $\pm$ SD = 22.3 $\pm$ 2.1 years) who did not participate in Experiment 1 and 2 participated in Experiment 3 and received compensation for their participation (cf. Experiment 1 for other details). They all gave written informed consent before the experiment.

### Stimuli
To observe the effect of affective priming that was triggered by the combination of valence and arousal, 60 neutral nouns (e.g., *rule*, *desk*) were selected from the same database as in Experiment 1 according to the same criteria. We considered words with valence values that ranged from 4 to 6 and word with arousal values less than 5 as neutral words. The same emotional words were used in Experiment 3 as in Experiment 1. They shared similar concreteness ($F_{4,174} = .18$, $p = .95$), word frequency ($F_{4,174} = 1.92$, $p = .11$), and character strokes ($F_{4,174} = 1.48$, $p = .21$) but differed in valence ($F_{4,174} = 26.86$, $p < .001$) and arousal ($F_{4,174} = 19.68$, $p < .001$). In Experiment 3, half of the pairs were emotional primes-emotional targets; the other half were emotional primes-neutral targets. Different from Experiments 1 and 2, the target stimuli in the emotional incongruent conditions were neutral words (see the last row of Table 1).

The 240 word-word pairs comprised 120 emotionally congruent pairs (30 PL-PL, 30 PH-PH, 30 NL-NL, 30 NH-NH) and 120 emotionally incongruent pairs (30 PL-neutral targets, 30 PH-neutral targets, 30 NL-neutral targets, 30 NH-neutral targets). The 240 word-pseudoword pairs were the same as in Experiment 1.

### Task and procedure
The experimental procedure was the same as in Experiment 1.

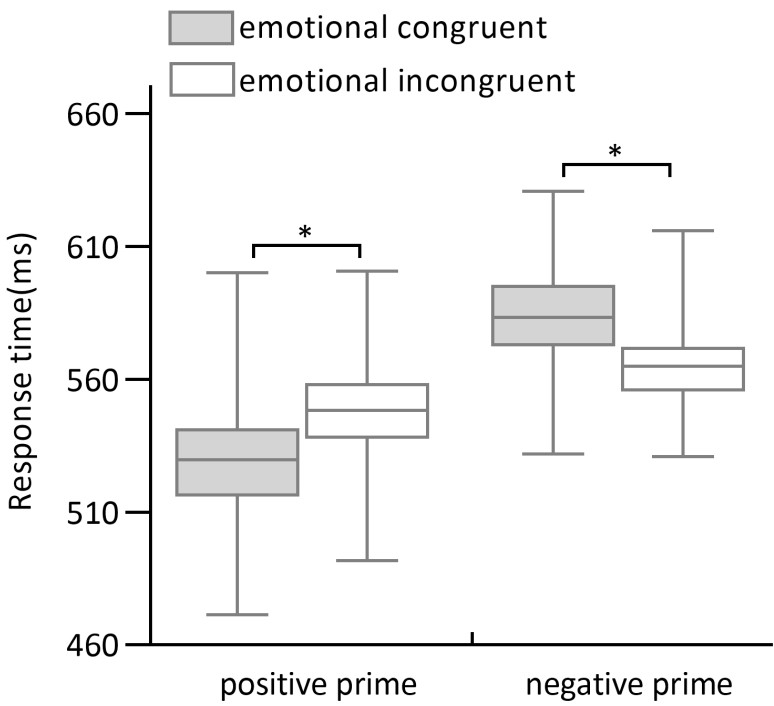

**Figure 3** **Response times of positive primes and negative primes in emotional congruence and incongruence conditions.** The gray column indicates the response times of prime-target emotional congruence condition and the gray slash indicates the response times of prime-target emotional incongruence condition.

## Results

Only correct response times were analyzed and reported, because overall accuracy was high (98.1%) and did not significant difference across all experimental conditions (range: 97.9–99.8%). Response times with 2.5 SDs above or below each participant's mean were excluded from analysis (.1% of the data). All calculations were conducted using an SPSS statistical package (version 18, SPSS inc., IBM company).

The 2 (emotional congruency of prime-target: congruent, incongruent) ×2 (prime valence: positive, negative) ×2 (prime arousal: high, low) repeated-measures ANOVA of response times revealed a significant main effect of prime valence ($F_{1,30} = 48.7$, $p < .001$, $\omega^2 = .61$; 538.7 ms for positive primes and 575.9 ms for negative primes). An interaction between prime valence and emotional congruency of the pairs was significant ($F_{1,30} = 59.82$, $p < .001$, $\omega^2 = .66$; Fig. 3). The simple-effect analysis showed that response times of incongruent trials (547.8 ± 4.1 ms) were significantly longer than of congruent trials (529.7 ± 5.0 ms; $F_{1,30} = 16.73$, $p < .001$, $d' = .57$) in positive primes, whereas response times of incongruent trials (565.2 ± 3.3 ms) were significantly shorter than of congruent trials in negative primes (585.5 ± 4.1 ms; $F_{1,30} = 55.58$, $p < .001$, $d' = .65$). No significant main effect of prime arousal was observed ($F_{1,30} = 3.1$, $p = .09$, $\omega^2 = .06$), with no other arousal-related effects (all $p > .05$).

## Discussion

Experiment 3 explored the ways in which a combination of valence and arousal of primes-targets modulate affective priming. No significant main effect of prime arousal and no interactions with prime arousal were observed. However, a significant main effect of prime valence was found, with a significant interaction between prime valence and emotional congruency. These findings are similar to Experiment 1 and suggest that prime valence rather than prime arousal influences the subsequent processing of target words when the emotionality of the prime-target varies with valence and arousal. Positive words as primes yielded a significant effect of affective priming, in which emotionally congruent trials were associated with shorter response times than incongruent trials. Negative words as primes yielded an opposite effect of affective priming, in which emotionally incongruent trials were associated with shorter response times than congruent trials. These findings are consistent with previous studies and provide additional evidence that the direction of priming effects is different between positive and negative prime words (*Pan et al., 2016*; *Kissler & Koessler, 2011*; *Rossell & Nobre, 2004*). The results of Experiment 3 can be explained by the affect-as-information approach (*Clore & Storbeck, 2006*), the density hypothesis (*Unkelbach et al., 2008*; *Koch et al., 2016*) and the automatic vigilance hypothesis (*Estes & Adelman, 2008*).

## GENERAL DISCUSSION

The present study investigated whether valence and arousal play equal roles in the subsequent processing of target words. We manipulated the relationship between primes and targets in terms of valence and arousal in a lexical decision-priming task. By comparing three types of priming effects that were triggered by pure valence information (Experiment 1), pure arousal information (Experiment 2), or a combination of the two (Experiment 3), we found an effect of valence on affective priming was obvious regardless of whether the relationship of the prime-target varied with valence, arousal, or the combination of valence and arousal. In contrast, an effect of arousal on affective priming only appeared in the condition in which the relationship of the prime-target varied with arousal level. Moreover, the valence-driven priming effect, arousal-driven priming effect, and emotional-driven priming effect were modulated by valence type but not by arousal level of word stimuli. These findings suggested that valence might a more robust and stable influence on affective priming compared with arousal, at least in the present lexical decision-priming task.

### Priming effects of valence, arousal, and a combination of valence and arousal

Our findings indicated a significant valence-driven priming effect in Experiment 1 and a significant arousal-driven priming effect in Experiment 2. In Experiment 3, although the emotional-driven priming effect did not reach statistical significance, a significant priming effect was found for positive primes, with an opposite effect for negative primes. These results indicated that affective priming effect was modulated by primed affective cues included valence, arousal and or a combination of the two. These findings are consistent with previous studies that demonstrated that responses to congruent pairs are significantly

faster than responses to incongruent pairs when the emotional dimension is primed in a lexical decision task (e.g., *Kissler & Koessler, 2011*; *Yao & Wang, 2013*; *Yao & Wang, 2014*; *Hinojosa, Méndez-Bértolo & Pozo, 2012*). The priming effects of valence, arousal, and emotionality can be explained by spreading activation within semantic networks. When the prime valence and/or arousal is evaluated implicitly, affectively congruent targets are more accessible, and participants can respond more easily relative to affectively incongruent targets (*Fazio, 2001*; *Klauer & Musch, 2003*).

However, our findings are seemingly inconsistent with the results of a noteworthy meta-analysis (*Herring et al., 2013*). The meta-analysis covered 25 years of affective priming studies ($k = 125$) and suggest that significant affective priming occurs in pronunciation (or naming) and evaluative decision tasks but not in lexical decision task. However, as mentioned by *Herring et al. (2013)*, regard to the priming effect of the lexical decision task should be taken with more caution due to the fact that their meta-analysis only included six effect sizes (from five experiments and four publications) that employed the lexical decision task, whereas it included 37 effect sizes (from 31 experiments and 20 publications) that employed the pronunciation/naming task. The smaller number of effect sizes makes it difficult to examine variables that moderate affective priming in the lexical decision task.

## Arousal priming effect only appeared under specific conditions

In Experiment 2, we manipulated the relationship of primes-targets based on arousal, and found a significant arousal-driven priming effect only in positive word pairs. The participants responded faster in arousal-congruent trials than in arousal-incongruent trials when the prime-target pairs were positive valence, which suggest that positive primes with high- or low-arousal can facilitate the processing of related positive targets, leading to differences in the activation of related arousal nodes between congruent and incongruent conditions. We infer possible reason for this result is that the arousal cue of a prime has the ability but alone is insufficient to activate all other concepts of the same arousal in semantic network, and thus has to need a support of positive valence. That is, spreading activation across an associative network of interconnected arousal nodes may be increased by positive information, arousal with a help of positive valence can easily capture a viewer's attention and increase cognitive resources during stimulus processing, and thus accelerate the semantic processing of target words.

In fact, with regard to arousal priming effect, there has no consistent conclusion. Some studies reported a significant arousal priming effect (e.g., *Zhang, Kong & Jiang, 2012*), other studies indicated no arousal priming effect or such effect only occurred under specific conditions (e.g., *Herring et al., 2015*; *Hinojosa et al., 2009*; *Hinojosa, Méndez-Bértolo & Pozo, 2012*). For example, Hinojosa and colleague (*2009*; *2012*) manipulated the arousal level of positive primes and targets and found a significant arousal priming effect at the electrophysiological level but not at behavioral level in an arousal categorization task. This finding suggest that the arousal information of prime stimuli have the capability to automatically pre-activating arousal-congruent targets by spreading activation within a semantic network, but this capability may relatively weaker and become visible by measuring event-related potentials with high temporal resolution.

## Stable but asymmetric effect of valence on affective priming

In all three of our experiments, we found that the priming effects of valence, arousal, and emotionality were modulated by valence. A standard priming effect (i.e., faster responses in congruent trials) was observed in the positive prime condition in all three experiments. An opposite priming effect (i.e., faster responses in incongruent trials) was observed in the negative prime condition in Experiments 1 and 3. No priming effect was found in the negative rather than positive prime condition in Experiment 2. These results indicated regardless of the relationship of primes-targets based on valence, arousal, or a combination of the two, the effect of valence on affective priming is obvious and stable, and positive and negative primes play either a facilitative or inhibitory role in affective priming. These results support previous findings with regard to the direction of priming effects differs between positive and negative primes (*Rossell & Nobre, 2004*; *Pan et al., 2016*), and further suggest that this difference is not varied with affective relation of primes-targets.

Specifically, our results showed that for positive primes, shorter latencies for congruent than for incongruent prime-target pairs were observed in all three experiments. The encoding perspective holds that all emotional information can be represented as nodes in semantic memory. Primes activate associations in memory that make the valence of targets more accessible, thus facilitating affective priming (*Fazio, 2001*). Positive information of primes can stably increase the accessibility and use of associations of prime-target pairs, which makes it easier for spreading activation between connected nodes, thus facilitating positive target encoding (*Clore & Storbeck, 2006*). According to the density hypothesis, another possible explanation is that positive information is more similar to other positive information, and thus an obvious advantage in spreading activation from positive primes to positive targets (*Unkelbach et al., 2008*; *Koch et al., 2016*).

For negative primes, opposite priming effects were found in Experiment 1 and 3, in which the affective relationship of primes-targets involved valence and the combine of valence and arousal, respectively. By contrast, such effect was not observed in Experiment 2, in which the affective relationship based on arousal level of primes-targets. These results indicate that opposite priming effects might occur under negative prime conditions, but a precondition is that the affective relationship between primes and targets should be based on valence or at least involve valence information. Based on the different organization of positive and negative information in memory, the opposite priming effects may be explained by the spreading inhibition hypothesis (*Clore & Storbeck, 2006*) or automatic vigilance hypothesis of emotion (*Estes & Adelman, 2008*). Because of prolonged attention to negative words, the processing of negative prime-negative target trials may produce a "double negative delay," which hinders the spread of negatively-valenced associations and reduces the optimization of word-processing. Consequently, negative targets are more slowly activated compared with positive and neutral targets when they are presented in negative primes. The opposite priming effects for negative primes that were observed in the present study supported and extended the automatic vigilance hypothesis and suggested that prolonged attention to negative words occurred when they were presented in isolation and also occurred when they were presented in the priming experiment.

Another interpretation of the present findings is based on the double-check hypothesis (*Aguado et al., 2018*), which states that the processing of negative stimuli involves a "double check" in terms of both valence and specific emotional content (e.g., anger, sadness, fear, and disgust). Thus, participants more easily judged negative stimuli in a positive context than negative stimuli in a negative context. For example, *Aguado et al. (2018)* used target faces that expressed happiness, fear, or anger that were presented after the participant had read a sentence that described anger, fear, or happiness-inducing events. The results in a congruency judgment task suggested that happy faces were recognized faster in happy contexts than in negative contexts, whereas angry faces were recognized faster in happy contexts than in negative contexts. In terms of the double-check hypothesis, recognizing a happy face requires only a valence check, whereas judging a negative face requires checks of both valence and emotion. In the present study, negative words were mainly selected based on their emotion dimension (*Russell, 2003*), without considering the specific emotional content of these words (e.g., anger, sadness, fear, and disgust). As a result, the opposite or null priming effects of negative primes may be explained by the fact that the encoding and processing of negative information is not only based on their valence but also requires the additional check of their specific emotional content.

## CONCLUSIONS AND LIMITATIONS

In summary, we found that valence exerts a differential and stronger effect than arousal on the subsequent processing of target words. The effect of valence on affective priming was obvious in all three priming conditions regardless of whatever the affective relationship of the prime-target varied with valence, arousal, or the combination of valence and arousal, whereas the effect of arousal on affective priming only appeared when the affective relationship based on arousal of primes-targets. Moreover, valence, arousal, and emotional priming effects were modulated by valence type, but not by arousal level. These findings are important because they replicate previous studies that suggested that valence and arousal both influence affective priming and extended this literature by systematically manipulating the relationship of primes-targets along the valence and arousal, indicating that the valence information of emotion words tended to be more stable than arousal information within the semantic system, at least in the present lexical decision-priming task.

One important limitation of the present study is that only a lexical decision-priming task and behavioral measure were employed, which may limit the generalization of our findings to the affective priming literature. According to a meta-analysis of nearly 30 years of affective priming research (*Herring et al., 2013*), the relative effects of valence and arousal on affective priming should be further examined by employing other non-evaluative (e.g., pronunciation/naming) or evaluative (e.g., valence/arousal decision) priming tasks and manipulating feature-specific attention allocation. Another limitation is that although the conclusions of the present study were drawn by comparing the priming effects of valence, arousal, and the combination of valence and arousal on subsequent target processing, the U-shaped relationship of valence and arousal ratings (e.g., *Yao et al., 2017*) may indicate that the arousal difference between high and low arousal words was less than the valence

difference between positive and negative words. Therefore, future studies should utilize more direct measurement methods, such as event-related potentials, that can distinguish subtle differences in the affective relationship between primes and targets in terms of valence and arousal to directly explore and compare brain electric activity that is evoked by information about the valence and arousal of emotional words.

### Funding

This research was supported by grants from the Nature Science Foundation of China (grant# no. 31600885) to Zhao Yao, and also was supported by the Fundamental Research Funds for the Central Universities and the Higher Education Reform Project of Xidian university to Zhao Yao. The funders had no role in study design, data collection and analysis, decision to publish, or preparation of the manuscript.

### Grant Disclosures

The following grant information was disclosed by the authors:
Nature Science Foundation of China: 31600885.
Fundamental Research Funds for the Central Universities: 20101196146, 20101194941, 20106186449.
Higher Education Reform Project of Xidian university: 20901190006.

### Competing Interests

The authors declare there are no competing interests.

### Author Contributions

- Zhao Yao conceived and designed the experiments, performed the experiments, analyzed the data, contributed reagents/materials/analysis tools, prepared figures and/or tables, authored or reviewed drafts of the paper, approved the final draft, prepare the manuscript.
- Xiangru Zhu conceived and designed the experiments, performed the experiments, analyzed the data, contributed reagents/materials/analysis tools, approved the final draft.
- Wenbo Luo conceived and designed the experiments, authored or reviewed drafts of the paper.

### Human Ethics

The following information was supplied relating to ethical approvals (i.e., approving body and any reference numbers):

The study was approved by the local Ethics Committee of Henan University (HUSON-2018-102). Each participant signed a written informed consent form prior to participation.

### Data Availability

The raw data are available as a Supplemental File.

## Supplemental Information

Supplemental information for this article can be found online at http://dx.doi.org/10.7717/peerj.7777#supplemental-information.

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
