# Peer review of "Valence makes a stronger contribution than arousal to affective priming"

_PeerJ, doi:10.7717/peerj.7777_

## Round 0.1 · original submission · Minor Revisions

I have been fortunate to receive reviews of your article by two experts in your field. Both reviewers were highly positive in their assessment of your study and article, as you can see from their feedback. Both suggest revisions, but I believe that if you can carefully respond to each of their points, your article will be suitable for publication in PeerJ.

I wish to note that reviewer 1 requests you run an additional study (i.e. collect more data). While I agree that this would enhance your report and give greater depth to your findings I do not believe that it is essential for acceptance into PeerJ. Therefore, if you decide not to collect such additional data, please provide your reasoning to reviewer 2 as you prepare your response. Perhaps such a study could be described as a future direction in your discussion.

·

Basic reporting

This work is nicely grounded in the antecedent literature and was thoughtfully controlled for many possible confounds as indicated by previous work. I was able to easily recapitulate all three analyses in JASP, and I appreciated the clear and concise descriptions of the relatively complex designs.

Experimental design

The present manuscript describes three affective priming experiments independently manipulating aspects of valence and arousal of the primes and targets in a lexical decision task, with a blocked design (where every block began with the same class of prime word, e.g. positive high-arousal). Pseudowords comprised half of the experimental materials and only ever occurred in the target position. The first experiment held the arousal level of the non-pseudoword prime-target pairs constant (at high or low values) and manipulated the positive/negative valence of the prime and target in a fully crossed design, nominally finding a facilitative effect of valence congruity given positive primes and an inhibitory effect of valence congruity given negative primes (though see #1 in the "Validity of Findings" section). The second experiment held the valence of the non-pseudoword prime-target pairs constant (at positive or negative valence) and manipulated the high/low arousal of the prime and target in a fully crossed design, again showing slower responses to negative trials but additionally showing an arousal congruity effect for positive pairs only (with incongruous trials slower than congruous trials). The third experiment mirrored the valence-priming design of experiment 1, but replaced all valence-incongruous targets with neutral words, again finding that negative targets elicited slow responses and additionally that the same neutral targets were responded to more slowly when preceded by a negative prime than a positive prime. The authors conclude that valence is a stronger contributor to the automatic processes elicited during reading, as both valence effects and valence-congruity effects were robustly observed at every opportunity to find one in the present experiments.
The number of repetitions was kept relatively low (the paper says 5, though see below), and word properties were matched across visual (e.g. # of strokes in the characters), lexical (frequency) and conceptual (concreteness, very important to control in this type of research) features.Though there are a few minor aspects of the analysis that could be improved (see below) and some instances of unclear text, possible typos, or minor omissions that could be cleaned up, the experiments are thoughtfully designed and competently executed.
1. I recommend using omega-squared as the effect size estimate instead of partial eta squared for samples of this size in order to correct for the optimistic bias almost certainly present here (see, for example, Okada 2013 in Behaviormetrika).
2. It’s not clear what the two values are on line 188, particularly since they extent beyond the lowest possible rating. These relatedness value could be more clearly summarized – perhaps a range of items-level proportions of “1” ratings would make sense? Or a cutoff threshold for the proportion of non-1 ratings?
3. Results section starting on line 214 should state that a repeated-measures ANOVA was used. If possible, however, a mixed-effects regression approach is likely better suited than an ANOVA, given the bounded set of possible prime-target pairs presented repeatedly to each participant. The present analysis can be thought of as assuming a representative sample of participants from which population inferences can be drawn. Also of interest to readers is the assumption that the particular items used are also representative, from which inferences about language processing more generally may also be drawn. The random effects of subjects is presently reflected in the analysis, but the random effects of items is not. See Barr et al 2014 in JML (the supplemental materials contains an extensive discussion related these regression models to ANOVAs).
4. Line 183 states the emotional words were repeated 5 times (3 in the prime position and 2 in the target position) but I’m counting 6? For example, the 30 PL words in experiment 1 appear twice in PL-PL trials (once in the prime and once in the target position), once in PL-NL trials, once in NL-PL trials, and twice in PL-pseudo trials. Could these numbers be double-checked and/or the relevant text about the trial composition clarified?
5. In the description of study materials, please state the cutoffs used for the different valence and arousal categories (including the “neutral” category).
6. Please provide part of speech numbers in the table of descriptive statistics for the study materials.
7. Please provide 95% confidence intervals in the figures. I also don’t recommend bar plots in cases where the low range of responses are implausible and/or artifactual (e.g. response times less than 100ms). A swarm plot or boxplot may be more appropriate for these data.

Validity of the findings

I find the interpretation somewhat un-parsimonious. Blocking by prime type, and centering the factorial analysis and follow-up simple effects on the prime valences (rather than target valences, as is common in the other behavioral research and ubiquitous in the relevant ERP research) somewhat obscures the fact that the primary valence and valence congruity results in the present experiments could largely be accounted for by a slowing of lexical decisions to negative targets. Addressing this alternative explanation is necessary for the present manuscript, as this alternative explanation does not actually require priming at all (though the arousal congruity effects are not subject to this possible alternative and are fine as is).
1. The block-based design implemented here very nicely reflects some of the observations made in the Herring et al metanalysis on experimental factors moderating affective priming effects, setting up a nice opportunity to make some meaningful contributions. However, the focus on the prime position here can make it difficult to parse some of the resulting effects given the literature on unprimed responses of those word classes. For example, the present paper reports a classic affective priming facilitation given a positive prime in all 3 experiments and a reverse valence priming effect (where congruous trials were slower) given a negative prime in two experiments. But this set of simple effects is also equivalent to simply concluding that “negative targets elicit slower lexical decision responses than positive targets”, a framing that parsimoniously accounts for the present data and also has some support in the literature (see for example Estes and Adelman 2008 in Emotion, and also Larsen et al 2008 in Emotion). This possibility is especially important in the present case because A) the primes are not task relevant (strategic participants would ignore them, and while the evidence here for prime-driven effects is sound, this design aspect may increase the importance of target-driven effects) and B) because of the blocking, the emotion aspects of the primes is perfectly predictable given a few trials exposure to the block. I’m actually preferentially sympathetic to the affective priming interpretation, but this view of the results (and design of the blocking) does carry this important consideration, and the issue of effects of the words that participants are making responses to needs to be addressed. I highly recommend a supplemental experiment showing lexical decision reaction times to the same positive and negative targets used in these experiments. If these targets show no difference in lexical decision times when presented in isolation, but do show differences in the priming experiment, it would be important evidence that the direction of the valence congruity effect may depend on numerous contextual factors (such as valence, arousal, blocking, and task) when the present results are interpreted in light of the broader literature. Without such an experiment, it’s unclear why the less parsimonious interpretation would be preferred.
2. The arousal difference between nominally high and low arousal words was significantly smaller than the valence difference between positive and negative words. While articulating the conclusions about the relative “strength” of valence and arousal priming effects, I would recommend addressing the possibility that these may have been influenced by the strength of the manipulations.
3. I think the reaction times to the pseudowords given different primes could be interesting. I recommend reporting these (in supplementary materials would be fine if the manuscript is desired to remain concise). There’s a possible concern here that deriving the pseudowords from the experimental materials could have left some residual possibility of activating at least the orthographic recognition of the emotional words, but confounds in the lexical decisions could be ruled out by assessing the marginal effects of the word type (e.g. PL, PH etc) each pseudoword was derived from.

Additional comments

Thank you for the thoughtful and concise manuscript. It was easy to read, addressed intuitive issues, and didn't get lost in the weeds. I rarely recommend collecting additional data, but in this case a very small additional experiment (of lexical decisions to these emotional words in isolation) bears significant implications to how the present results should best be interpreted, and I think would considerably strengthen the arguments in the paper. All my other points were minor. I look forward to seeing this manuscript published after revisions.

Reviewer 2 ·

Basic reporting

no comment

Experimental design

no comment

Validity of the findings

no comment

Additional comments

I’ve reviewed this paper already and thought it should have been published the first time. I’ve looked back at my old review and have found that the author’s addressed many of my original concerns. Below are some areas the authors could improve upon. Like I said to the authors over a year ago, the authors are commended for their systematic investigation of valence and arousal contributions on evaluative priming. Unkelbach has been working to update priming accounts that more clearly involve the hedonic valence and this study is noteworthy to this end.

Primary concerns

-I think the authors are sill diving into the affective priming paradigm too quickly. Why not add a little pizzazz to the very beginning by talking about why anyone would be interested in indirect measures in the first place?

-Here were my comments from a year ago that the author’s did well in addressing:

“…On the same page it seems the rationale that is provided is that LDT allows for elucidating purely valence or arousal dimensions given it doesn’t provide a particular mindset of either valence or arousal. But, given an attentional sensitization account (see Spruyt’s work), it does not appear that a strong prediction of affective priming can be made using an LDT provide the participant is in more of a semantic mindset. I encourage the author’s to spell out the rationale more clearly for using the LDT and pit it against Spruyt and Kiefer’s work. Similarly, an addition to the discussion (p. 19, para 1) is in order to compare the current results to a recent meta-analysis showing null effects of the LDT.”

Is that accurate the LDT creates a relative neutral and not a semantic mindset? At any rate, I still wrestle a bit with the fact that one wouldn’t then anticipate any behavioral effect given the Herring meta-analysis. This might be a nice spot to mention where using ERPs in this area is particularly well suited. That is, we know the evidence for evaluative priming is weak for the LDT behaviorally, but ERPs offer an opportunity to investigate whether arousal or valence effects are stronger.

-Table 1: please provide superscripts to where the four words types (i.e., PH, NH, PL, NL) differ in valence and arousal.

Minor concerns

-effect size estimates are missing in Experiment 3

---

## Round 0.2 · Minor Revisions

The two reviewers who reviewed your original submission have now reviewed the revised version of your article. Both praised you for responding so thoroughly to their feedback. I concur with both reviewers that your article is suitable for publication in PeerJ.

However, as Reviewer 1 also notes, there are a few areas of your manuscript that lack clarity and would benefit from being proof read one more time. I have read through your article and made a few suggested edits. You can see them on the attached annotated manuscript. I suggest that you make these changes and complete one final careful revision of your article before resubmission.

If you are able to make these final few minor edits for language clarity, it will be my pleasure to accept your article without further review.

·

Basic reporting

- New figures look great.
- Was able to reproduce all analyses with the shared data of all 3 experiments during my previous review.
- Recommend running the manuscript by a copy editor to better align the professional English with the style of the other publications in this journal.

Experimental design

- All typos and omissions noted in the previous review have been addressed.

Validity of the findings

- Authors graciously included additional data and analyses of interest in the supplemental materials after the last round of reviews.
- Experiment is well controlled and precisely described.
- The discussion of the possibility that the arousal manipulation was not as "strong" as the valence manipulation could be made much simpler. The U-shaped relationship between explicit valence and arousal ratings seems a bit tangential to the primary concern on this point, as does the temporal resolution of ERPs. It seems fine to simply present the limitation, perhaps noting that future attempts could be made to achieve valence and arousal manipulations of similar magnitudes.

Additional comments

- I appreciate the authors' considered edits and gracious response to reviewer comments.
- Some of the intro and discussion, particularly the newer sections that likely haven't had the benefit of quite so many editing passes, occasionally structured information in ways that were hard to follow. Specifically, it was not always clear what argument was being served by a reference to concepts or findings from the broader literature. I recommend one more pass to check that each reference isn't just topical, but connects to a clearly articulated argument. And additionally, that those arguments have a logic that is transparent and well-structured.

Reviewer 2 ·

Basic reporting

meets standards

Experimental design

meets standards

Validity of the findings

meets standards

Additional comments

The authors did an excellent job addressing my concerns. I have no further comments.

---

## Round 0.3 · accepted · Accept

Thank you for responding to mine and the reviewer's final comments and suggested edits. It is my pleasure to accept your article for publication.